# Activity-dependent death of transient Cajal-Retzius neurons is required for functional cortical wiring

Martina Riva[1,2,3†], Ioana Genescu[4†], Chloé Habermacher[3,5†], David Orduz[5§], Fanny Ledonne[2], Filippo M Rijli[6], Guillermina López-Bendito[7], Eva Coppola[1,2,3], Sonia Garel[4‡*], Maria Cecilia Angulo[3,5‡*], Alessandra Pierani[1,2,3‡*]

[1]Institut Imagine, Université de Paris, Paris, France; [2]Institut Jacques Monod, CNRS UMR 7592, Université de Paris, Paris, France; [3]Institute of Psychiatry and Neuroscience of Paris (IPNP), INSERM U1266, Université de Paris, Paris, France; [4]Institut de Biologie de l'École Normale Supérieure (IBENS), Département de Biologie, École Normale Supérieure, CNRS, INSERM, Université PSL, Paris, France; [5]INSERM U1128, Paris, France; [6]Friedrich Miescher Institute for Biomedical Research, Basel, Switzerland; [7]Instituto de Neurociencias de Alicante, Universidad Miguel Hernandez, Sant Joan d'Alacant, Spain

*For correspondence:
garel@biologie.ens.fr (SG);
maria-cecilia.angulo@
parisdescartes.fr (MCA);
alessandra.pierani@inserm.fr (AP)

†These authors contributed
equally to this work
‡These authors also contributed
equally to this work

Present address: §Gfi
informatique, Saint-Ouen-sur-
Seine, France

Reviewing editor: Carol A
Mason, Columbia University,
United States

**Abstract** Programmed cell death and early activity contribute to the emergence of functional cortical circuits. While most neuronal populations are scaled-down by death, some subpopulations are entirely eliminated, raising the question of the importance of such demise for cortical wiring. Here, we addressed this issue by focusing on Cajal-Retzius neurons (CRs), key players in cortical development that are eliminated in postnatal mice in part via Bax-dependent apoptosis. Using Bax-conditional mutants and CR hyperpolarization, we show that the survival of electrically active subsets of CRs triggers an increase in both dendrite complexity and spine density of upper layer pyramidal neurons, leading to an excitation/inhibition imbalance. The survival of these CRs is induced by hyperpolarization, highlighting an interplay between early activity and neuronal elimination. Taken together, our study reveals a novel activity-dependent programmed cell death process required for the removal of transient immature neurons and the proper wiring of functional cortical circuits.

## Introduction

An emerging player in the assembly of neuronal networks is programmed cell death (PCD). In the nervous system, programmed cell death (PCD) fine-tunes the density of neuronal populations by eliminating 20–40% of overproduced neurons (*Fuchs and Steller, 2011*; *Causeret et al., 2018*; *Wong and Marín, 2019*). Only few populations of the mouse cerebral cortex almost completely disappear during the first two postnatal weeks. Amongst these, Cajal-Retzius cells (CRs), the first-born cortical neurons lying in the superficial Layer I (LI), undergo extensive cell death in the mouse during the second postnatal week (*Ledonne et al., 2016*). The persistence of CRs during postnatal life is increased in malformations of cortical development (MCDs) and epilepsies thereby opening the intriguing possibility that the maintenance of CRs contributes to the dysfunction of cortical circuits (for review see *Luhmann, 2013*).

CR play pivotal roles at multiple steps of early cortical development, in addition to their best-known role in the control of radial migration (*Ishii et al., 2016*). They comprise three molecularly distinct subtypes which migrate from different sources that surround the cortical primordium: (i) septum

and eminentia thalami-derived CRs of the ΔNp73/Dbx1 lineage (SE-CRs); (ii) hem-derived CRs of the ΔNp73/Wnt3a lineage (hem-CRs); (iii) pallial-subpallial boundary-derived CRs of the Dbx1 lineage (PSB-CRs) (*Bielle et al., 2005*; *Yoshida et al., 2006*; *Tissir et al., 2009*). Our previous work revealed that their specific embryonic distribution in distinct territories plays key functions in wiring of cortical circuits by controlling the size of functional areas, both primary and higher-order, as well as targeting of thalamocortical afferents (*Griveau et al., 2010*; *Barber et al., 2015*; *Barber and Pierani, 2016*). More recently, we showed that subtype-specific differences also exist in their elimination during early postnatal life, with SE- but not hem-derived CRs dying in a Bax-dependent manner, which is a critical player of the apoptotic pathway (*Ledonne et al., 2016*).

Before their disappearance, CRs express ionotropic glutamatergic and GABAergic receptors and are embedded into immature circuits where they mainly receive GABAergic synaptic inputs, suggesting that these transient cells might have an activity-dependent role in the development of cortical networks (*Kirischuk et al., 2014*). Consistently, we found that CRs density shapes axonal and dendritic outgrowth in LI and impacts onto the excitation/inhibition (E/I) ratio in upper cortical layers (*de Frutos et al., 2016*). Conversely, activity was also proposed as one of the drive of CR demise. Studies from *Del Río et al. (1996)* first highlighted the role played by electrical activity on CR death in vitro. CRs were shown to display electrophysiological features of immature neurons (*Kirischuk et al., 2014*; *Barber and Pierani, 2016*), including the persistent depolarizing action of GABA which was suggested to depend on the maintenance of the chloride inward transporter NKCC1 and the absence of the outward transporter KCC2 (*Mienville, 1998*; *Achilles et al., 2007*; *Pozas et al., 2008*). Interestingly, pharmacological inhibition of activity and GABA signaling in vitro and global inactivation of NKCC1 in vivo were shown to reduce the death of CRs (*Blanquie et al., 2017a*; *Blanquie et al., 2017b*). However, very little is known on the role of electrical activity in CR subtype-specific death in vivo, as well as its contribution to the construction of functional and dysfunctional cortical circuits.

Here we show that hyperpolarization of CR neurons by Kir2.1-dependent expression prevented cell death of ΔNp73- but not Wnt3a-derived CRs, corresponding to SE-CRs. By comparing two different mutants in which CR death was similarly rescued, we found that abnormal SE-CR survival promotes exuberance of dendrites and spine density in pyramidal neurons in an activity-dependent manner. This results in an E/I imbalance due to an increase in the excitatory drive of upper cortical neurons. Our findings show that neuronal activity is involved in CR subtype-specific death and argue in favor of an unappreciated role of the disappearance of these transient neurons in controlling the morphology of pyramidal neurons and the functional properties of their cortical excitatory networks at postnatal stages.

## Results

### Subsets of CRs die in an activity-dependent manner

The ΔNp73[cre/+] mouse line targets approximately 80% of CRs, namely hem-CRs (Wnt3a lineage) and SE-CRs (*Tissir et al., 2009*; *Griveau et al., 2010*; *Yoshida et al., 2006*). ΔNp73-CRs, but not Wnt3a-derived hem-CRs, were shown to be eliminated postnatally via Bax-dependent death, indicating that SE-CRs undergo apoptosis. However, the trigger of such apoptosis or the mechanisms regulating the death of other CR populations are still unknown (*Ledonne et al., 2016*). To decipher whether activity might regulate the death of specific subsets of CRs in vivo, we first overexpressed the hyperpolarizing channel Kir2.1 (R26[Kir2.1mcherry/+]) (*Moreno-Juan et al., 2017*) using the ΔNp73[cre/+] line. Tracing of CRs at several postnatal stages in control and Kir2.1-expressing mice was performed using DsRed immunostainings to visualize tdTomato and mcherry reporters, respectively. We found that the density of CRs in the somatosensory barrel cortex was unchanged in these animals at postnatal day 7 (P7), that is before CRs undergo massive cell death (*Figure 1A and B*). In contrast, Kir2.1 channel overexpression in ΔNp73-CRs resulted in an increase of CRs with respect to controls in the somatosensory cortex at P15 and P25 (*Figure 1A and B*). Importantly, using whole-cell recording, we checked that rescued CRs were, as expected, hyperpolarized and displayed a decrease in the input resistance without exhibiting changes in action potential properties at both P15 and P25 (*Figure 1—figure supplement 1A-C*). Biocytin-filling and immunostaining further revealed that most morphological properties were preserved in Kir2.1-expressing CRs, apart from a reduced soma at

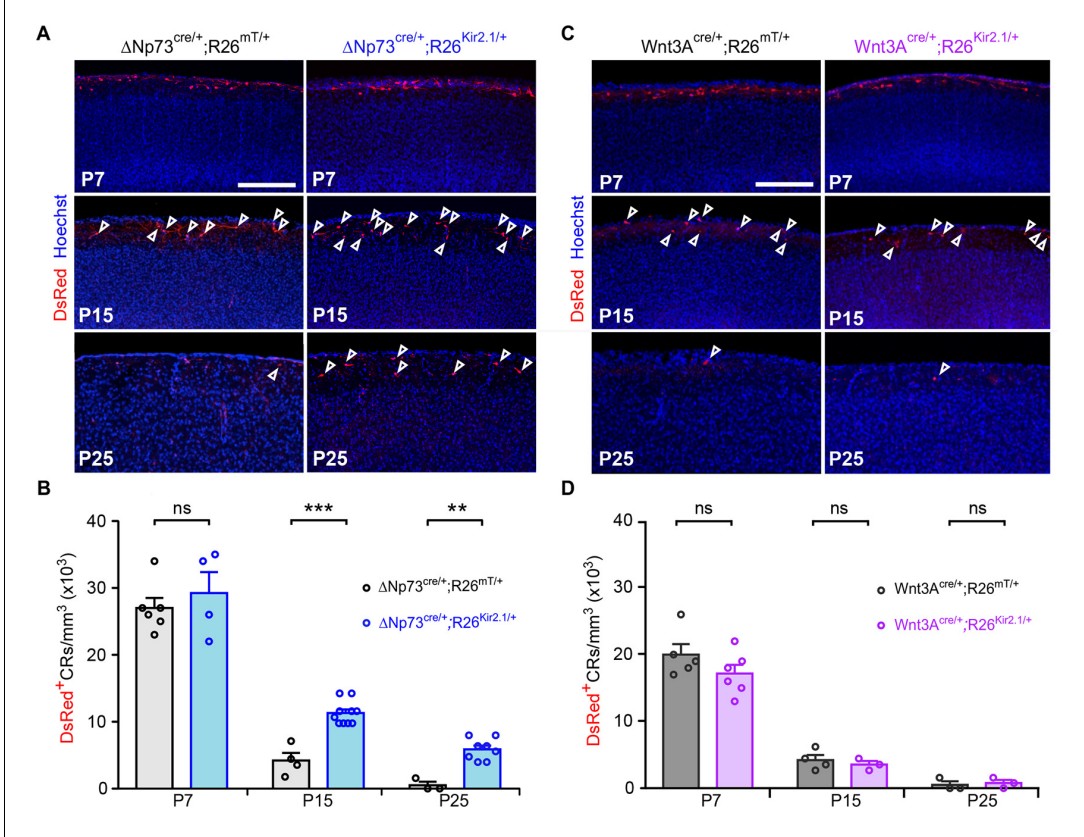

**Figure 1.** Hyperpolarization induces the survival of SE-CRs. (**A**) Confocal images of cortical sections from P7, P15 and P25 ΔNp73$^{cre/+}$;R26$^{mT/+}$ controls (left) and ΔNp73$^{cre/+}$;R26$^{Kir2.1/+}$ mutants (right) stained for DsRed (red) and Hoechst (blue). Arrowheads indicate DsRed$^+$ CRs at P15 and P25. (**B**) Quantification of CR density (CRs/mm$^3$) at the pial surface in the somatosensory (S1) cortex (P7: n = 6 for controls and n = 4 for mutants, p=0.716; P15: n = 4 for controls and n = 10 for mutants, p=0.001; P25: n = 3 for controls and n = 8 for mutants, p=0.006). (**C**) Confocal images of cortical sections from P7, P15 and P25 Wnt3a$^{cre/+}$;R26$^{mT/+}$ controls (left) and Wnt3a$^{cre/+}$;R26$^{Kir2.1/+}$ mutants (right) stained for DsRed (red) and Hoechst (blue). Arrowheads indicate DsRed$^+$ CRs at P15 and P25. For simplicity, arrowheads were not displayed at P7, as there are too many CRs at this stage. (**D**) Quantification of CR density (CRs/mm$^3$) at the pial surface in the somatosensory (S1) cortex (P7: n = 5 for controls and n = 6 for mutants, p=0.2251; P15: n = 4 for controls and n = 3 for mutants, p=0.771; P25: n = 3 for controls and n = 3 for mutants, p=0.813). Mann-Whitney U Test. Scale bar represents 200 μm. Data used for quantitative analyses as well as the numerical data that are represented in graphs are available in *Figure 1—figure supplement 1—source data 1*. The online version of this article includes the following source data and figure supplement(s) for figure 1:

**Figure supplement 1.** Electrophysiological properties and morphology of rescued CRs in ΔNp73$^{cre/+}$; R26$^{Kir2.1/+}$ mice.

**Figure supplement 1—source data 1.** Density and properties of CRs in the Kir2.1 model.

P25 (*Figure 1—figure supplement 1D*). In particular, rescued CRs displayed similar branching length in LI (*Figure 1—figure supplement 1B and E*) and co-expressed Reelin (Reln) from P7 to P25 (*Figure 1—figure supplement 1F and G*). Taken together, these experiments show that hyperpolarization of ΔNp73-CRs does not drastically alter cardinal morphological features of CRs but prevents their complete elimination.

A similar proportion of CRs were rescued either by hyperpolarization (*Figure 1A and B*) or by preventing apoptosis (*Ledonne et al., 2016*), raising the intriguing possibility that the same subpopulation of CRs, SE-CRs, might be preserved in both conditions. Since hem-CRs are not eliminated via a Bax-dependent process (*Ledonne et al., 2016*), we investigated whether their survival is sensitive to hyperpolarization. We thus overexpressed Kir2.1 specifically in hem-CRs using the Wnt3a$^{cre/+}$, which corresponds to about 70% of the ΔNp73-CRs at early postnatal stages (*Figure 1C and D*). We found that hem-CR death was unaffected in this mouse line (*Figure 1C and D*). Moreover, taking into account cortical growth, the proportions of rescued cells in the somatosensory cortex at P25 was evaluated to approximately 30% of the initial pool of ΔNp73-CRs, which corresponds to the expected number of SE-CRs (*Bielle et al., 2005*; *Yoshida et al., 2006*; *Tissir et al., 2009*).

Collectively, these results show that the death of a specific subset of ΔNp73$^{cre/+}$ SE-CRs is both Bax-dependent and activity-dependent.

## CRs rescued by hyperpolarization or blocking Bax-dependent apoptosis are integrated in neuronal circuits

It has been established that CRs are integrated in functional circuits early in the developing postnatal neocortex (*Kilb and Luhmann, 2001*; *Soda et al., 2003*; *Sava et al., 2010*; *Cocas et al., 2016*). CRs receive GABAergic synaptic inputs and, despite the expression of NMDA receptors (NMDARs) on CR membranes, the presence of NMDAR-mediated synaptic responses is still under debate (*Kilb and Luhmann, 2001*; *Soda et al., 2003*; *Sava et al., 2010*; *Schwartz et al., 1998*; *Mienville and Pesold, 1999*; *Radnikow et al., 2002*; *Anstötz et al., 2014*). To test whether CRs harbored functional GABAergic and/or glutamatergic synapses during the cell death period and after their rescue, we recorded spontaneous (sPSCs) and evoked postsynaptic currents (ePSCs) of fluorescent CRs with a KCl-based intracellular solution. First, at P9-11 in control ΔNp73$^{cre/+}$;R26$^{mt/+}$ mice, sPSCs sensitive to the GABA$_A$ receptor (GABA$_A$R) antagonist SR95531 (10 µM) were observed in CRs held at −60 mV, confirming that CRs are innervated mainly by functional GABAergic synaptic inputs (*Figure 2—figure supplement 1A–B*; data not shown for SR95531 application; n = 7). Low-frequency stimulations in LI easily elicited ePSCs that were also completely blocked by SR95531 application (*Figure 2—figure supplement 1C–D*), even in 0 mM Mg$^{2+}$ condition, which relieves the Mg$^{2+}$ block of NMDA receptors. Taken together, these results indicate that ePSCs were mediated by GABA$_A$Rs and not by AMPA or NMDA receptors. In agreement with previous studies (*Sun et al., 2019*), the presence of GABAergic synaptic inputs on CRs was confirmed by immunostainings against the presynaptic marker GAD65/67 and the postsynaptic marker Gephyrin (*Figure 2—figure supplement 1E*). Since the depolarizing action of GABA in CRs was proposed to be partly due to the lack of expression of KCC2, the chloride transporter responsible for maintaining a low intracellular chloride concentration (*Mienville, 1998*; *Achilles et al., 2007*; *Pozas et al., 2008*; *Blanquie et al., 2017a*; *Blanquie et al., 2017b*), we also checked for the protein expression of this transporter. In agreement with previous reports (*Achilles et al., 2007*; *Pozas et al., 2008*), control GFP$^+$ CRs in ΔNp73$^{cre/+}$;Tau$^{GFP/+}$ mice expressed very low to undetectable levels of KCC2 (*Figure 2—figure supplement 1F*). Thus, during the timeperiod of their activity-dependent death, CR cells receive solely GABAergic synaptic inputs.

We further explored whether these inputs are maintained at later stages, when CR cells are not eliminated. To this aim, we performed the same experiments at P23-28 in ΔNp73$^{cre/+}$;R26$^{Kir2.1/+}$ and ΔNp73$^{cre/+}$;Bax$^{lox/lox}$;R26$^{mT/+}$ mice (*Figure 2A–B*). First, we found that sPSCs in rescued CRs had similar frequencies, amplitudes and kinetics in both models (*Figure 2A–B*). Moreover, sPSCs were completely abolished by GABA$_A$R antagonist SR95531, indicating that rescued CRs remained innervated by functional GABAergic synaptic inputs (data not shown for SR95531 application, n = 8 and n = 5 for ΔNp73$^{cre/+}$;R26$^{Kir2.1/+}$ and ΔNp73$^{cre/+}$;Bax$^{lox/lox}$;R26$^{mT/+}$ mice, respectively). Consistently, the complete block of ePSCs by SR95531, even in 0 mM Mg$^{2+}$, revealed that ePSCs were exclusively mediated by GABA$_A$Rs at hyperpolarized holding potentials, as observed in younger mice (*Figure 2C–D* and *Figure 2—figure supplement 1C–D*). While these GABAergic inputs were preserved in rescued CRs, we observed a reduced sPSC frequency and ePSC amplitude compared to P9-P11 mice, suggesting a decreased CR connectivity in the more mature neocortex (*Figure 2A–B* and *Figure 2—figure supplement 1A–B*).

Altogether, these results indicate that rescued CRs, as previously showed in the early postnatal neocortex (*Soda et al., 2003*; *Sava et al., 2010*; *Kilb and Luhmann, 2001*), receive solely GABAergic synaptic inputs at the time of their death. They further demonstrate that rescued CRs in both mouse models, inspite of a reduced connectivity compared to earlier stages, are kept integrated in functional neuronal circuits.

## Survival of electrically-active CRs triggers dendritic exuberance of upper layer pyramidal neurons

In order to test whether CR aberrant survival may alter the function of other neurons in upper cortical layers, we used biocytin-filling and confocal 3D reconstructions to study the morphology of Layer II (LII) and LIII pyramidal neurons in the somatosensory barrel cortex of both ΔNp73$^{cre/+}$;R26$^{Kir2.1/+}$

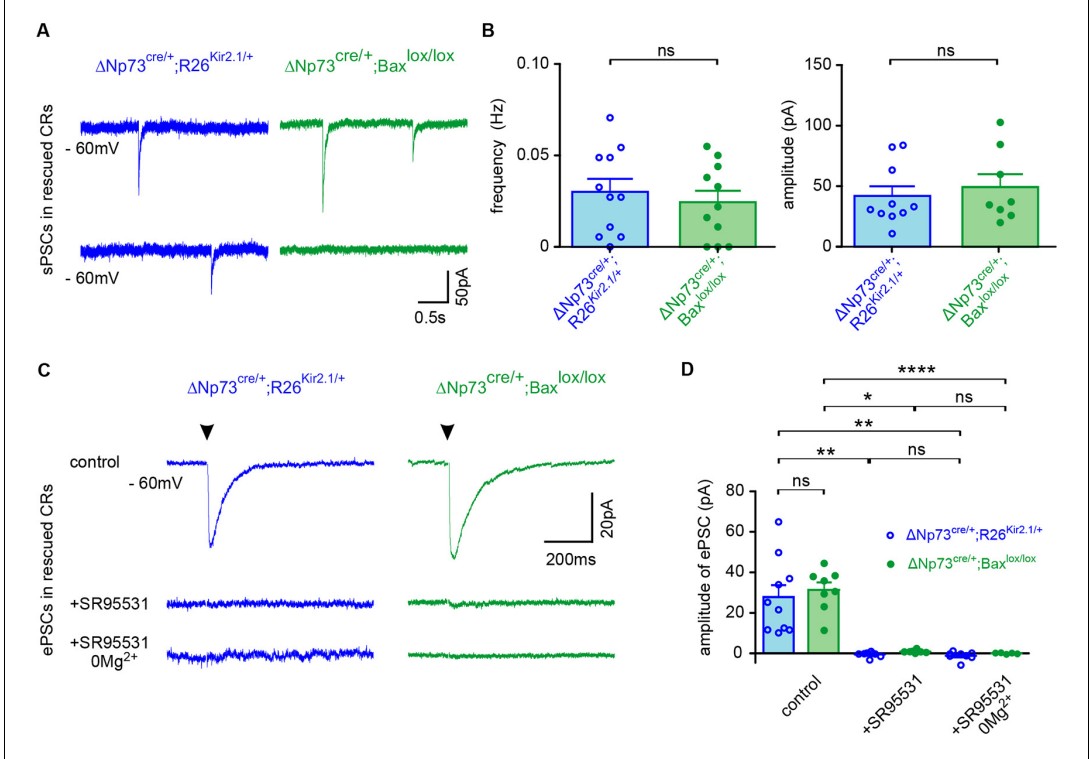

**Figure 2.** Pure GABAergic sPSCs and ePSCs in rescued CRs. (**A**) Spontaneous PSCs (sPSCs) recorded in rescued CRs from $\Delta Np73^{cre/+}$;$R26^{Kir2.1/+}$ at P27 (blue) and $\Delta Np73^{cre/+}$;$Bax^{lox/lox}$ mutants at P26 (green), respectively. (**B**) Plots of the frequency and amplitude of sPSCs (n = 11 for $\Delta Np73^{cre/+}$; $R26^{Kir2.1/+}$ and n = 11 for $\Delta Np73^{cre/+}$; $Bax^{lox/lox}$ mice at P24-29; frequency: p=0.552, amplitude: p=0.580, Student T Test). Rise time is 2.10 ± 0.42 ms vs 1.02 ± 0.20 ms and decay time 34.26 ± 6.39 ms vs 29.14 ± 3.56 ms for $\Delta Np73^{cre/+}$;$R26^{Kir2.1/+}$ and $\Delta Np73^{cre/+}$;$Bax^{lox/lox}$ mice, respectively. (**C**) Mean evoked PSCs (ePSCs) for rescued CRs respectively from a $\Delta Np73^{cre/+}$;$R26^{Kir2.1/+}$ mutant at P29 (blue) and a $\Delta Np73^{cre/+}$;$Bax^{lox/lox}$ mutant at P26 (green) upon stimulation of LI neuronal fibers (stimulation time, arrowhead) in control conditions (top), with SR95531 (middle) and SR95531 in $Mg^{2+}$-free solution (bottom). Note that ePSCs completely disappeared after bath application of SR95531. (**D**) Amplitudes of ePSCs in control conditions, with SR95531 and with SR95531 in $Mg^{2+}$-free solution ($\Delta Np73^{cre/+}$;$R26^{Kir2.1/+}$ mice at P24-29: $n_{control}$ = 10, $n_{SR95531}$ = 8 and $n_{SR95531}/_{Mg2+free}$=8; $\Delta Np73^{cre/+}$;$Bax^{lox/lox}$: $n_{control}$ = 8, $n_{SR95531}$ = 5 and $n_{SR95531}/_{Mg2+free}$=5; Kruskal-Wallis test followed by a Bonferroni multiple comparison when comparing the three conditions for each mutant; Student T test for comparison of control ePSCs between $\Delta Np73^{cre/+}$;$R26^{Kir2.1/+}$ and $\Delta Np73^{cre/+}$;$Bax^{lox/lox}$ mutants, p=0.638). To detect CRs in $\Delta Np73^{cre/+}$;$Bax^{lox/lox}$ mutants the $R26^{mT/+}$ reporter line was used. Data used for quantitative analyses as well as the numerical data that are represented in graphs are available in *Figure 2—figure supplement 1—source data 1*.

The online version of this article includes the following source data and figure supplement(s) for figure 2:

**Figure supplement 1.** Pure GABAergic sPSCs and ePSCs in control CRs during early postnatal development.

**Figure supplement 1—source data 1.** Evoked and Spontaneous PSCs in rescued and developing CRs.

and $\Delta Np73^{cre/+}$;$Bax^{lox/lox}$ mice. While no major changes were observed in the morphology of LII/III pyramidal neurons of $\Delta Np73^{cre/+}$;$R26^{Kir2.1/+}$ mice compared to their matched controls (***Figure 3— figure supplement 1***), major defects were observed for these neurons in $\Delta Np73^{cre/+}$;$Bax^{lox/lox}$ mutants (***Figure 3***). Quantitative analyses revealed an increase in the number of apical and basal dendritic branches at P25 in $\Delta Np73^{cre/+}$;$Bax^{lox/lox}$ compared to controls (***Figure 3B***). In order to further characterize cell complexity in relation with the distance from the soma, we performed a Sholl analysis. We observed that LII/III pyramidal cells displayed an increased complexity for apical dendrites (between 180 µm and 240 µm from the soma) as well as for basal dendrites (between 60 and 80 µm from the soma) (***Figure 3C***). Interestingly, no statistically different changes in the number of dendritic branches (***Figure 3—figure supplement 1B***) or the complexity of apical and basal dendrites (***Figure 3—figure supplement 1C***) were detected in LII/LIII pyramidal neurons of the $\Delta Np73^{cre/+}$;$R26^{Kir2.1/+}$ model, in which SE-CRs survive but are hyperpolarized. Thus, the survival of a specific subset of CRs has a general promoting impact onto the dendritic tree of upper layer pyramidal neurons only when they keep their normal intrinsic excitability. Collectively, our analyses show

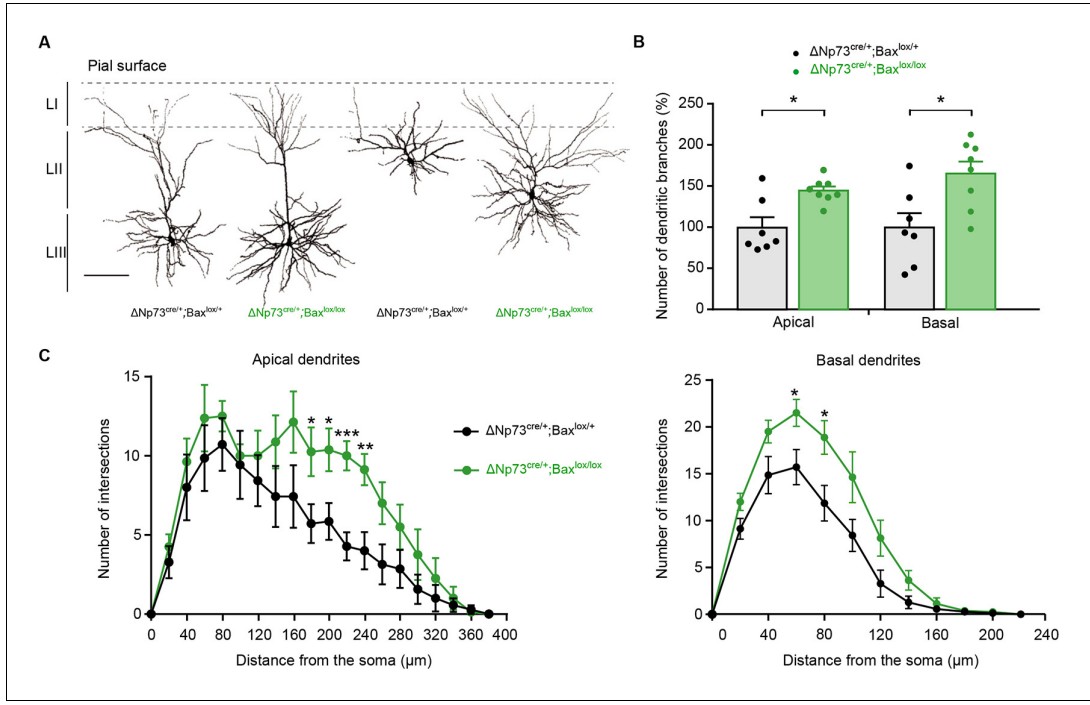

**Figure 3.** Increased dendritic branches in LII/LIII pyramidal neurons of $\Delta$Np73$^{cre/+}$;Bax$^{lox/lox}$ mutants. (**A**) Representative examples of LII/III pyramidal neurons filled with biocytin in control (P25) and $\Delta$Np73$^{cre/+}$;Bax$^{lox/lox}$ mutant (P24) somatosensory cortex. (**B**) Quantification of the number of dentritic branches in control and $\Delta$Np73$^{cre/+}$;Bax$^{lox/lox}$ mutant LII/III pyramidal neurons, expressed as a percentage of dendritic branches relative to the mean of controls (n = 7 for controls and n = 8 for mutants at P23-28 p=0.0182 for apical dendrites and p=0.014 for basal dendrites; Mann-Whitney U Test). (**C**) Sholl analysis for the apical and basal dendrites in control and $\Delta$Np73$^{cre/+}$;Bax$^{lox/lox}$ mutants showing an increased cell complexity between 180 and 240 $\mu$m (p-value=0.04, 0.027, 0.0007 and 0.005, respectively) and 60 and 80 $\mu$m (p value=0.027 and 0.019, respectively) from the soma, respectively (n = 7 for controls and n = 8 for mutants). Multiple T-test. Scale bar represents 100 $\mu$m. Data used for quantitative analyses as well as the numerical data that are represented in graphs are available in *Figure 3— figure supplement 1—source data 1*.

The online version of this article includes the following source data and figure supplement(s) for figure 3:

**Figure supplement 1.** Morphological reconstruction of LII/LIII pyramidal neurons *in* $\Delta$Np73$^{cre/+}$;R26$^{Kir2.1/+}$ mutants.
**Figure supplement 1—source data 1.** Morphological analyses of layer II/III pyramidal cells in the Bax and Kir2.1 models.

that SE-CRs survival in LI promotes an exuberance of apical and basal dendrites of LII/LIII pyramidal neurons in an activity-dependent manner.

## Survival of electrically-active CRs increases excitatory entries in upper layer pyramidal neurons

To examine whether the defects of the dendritic arborization of LII/III pyramidal neurons were related to changes in the synaptic inputs received by these neurons, we first examined spines on both apical and basal dendrites of biocytin-filled pyramidal cells (*Figure 4A–D*). Because excitatory synapses are formed on dendritic spines, the latter can be used as a proxy for the quantification of those synapses. For apical dendrites, we examined terminal ramifications in LI, whereas for basal dendrites, we considered horizontal branches approximately at the same distance from the soma. Spine density on both apical and basal dendrites of pyramidal cells was significantly increased in $\Delta$Np73$^{cre/+}$;Bax$^{lox/lox}$ mutants compared to controls (*Figure 4A and B*) whereas no differences were observed in $\Delta$Np73$^{cre/+}$;R26$^{Kir2.1/+}$ mutants (*Figure 4C and D*). These results indicate that the survival of electrically-active CRs not only triggers a dendritic exuberance, but also drives an increase in spine densities.

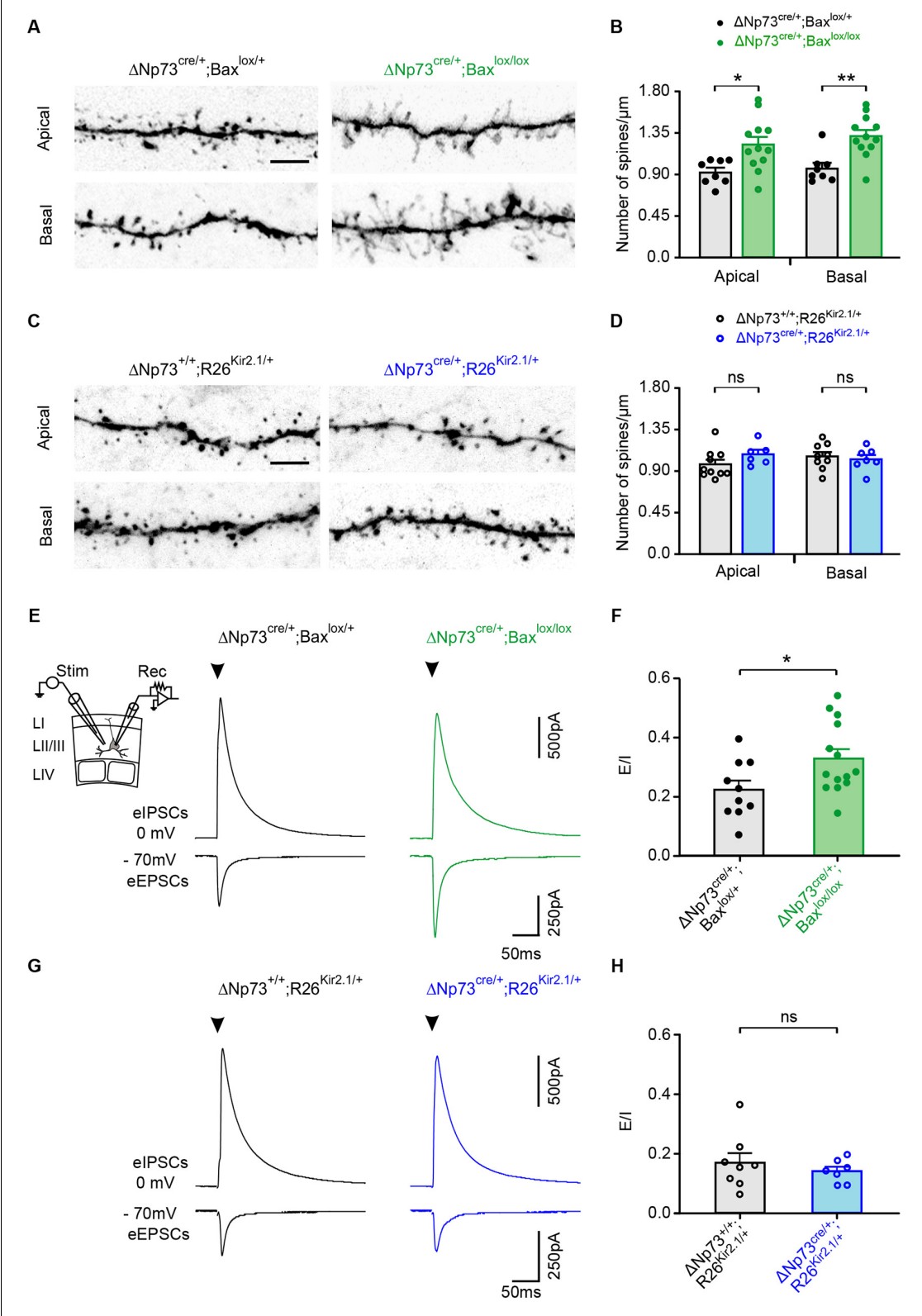

**Figure 4.** Spine density and evoked synaptic activity recorded in LII/LIII pyramidal neurons in both ΔNp73$^{cre/+}$;Bax$^{lox/lox}$ and ΔNp73$^{cre/+}$;R26$^{Kir2.1/+}$ mutants. (A, C) Representative confocal images showing spines in apical and basal dendritic segments of controls (left) and ΔNp73$^{cre/+}$;Bax$^{lox/lox}$ (A, right) and ΔNp73$^{cre/+}$;R26$^{Kir2.1/+}$ mutants (C, right) at P24-25. (B, D) Quantification of the spine density (number of spines/μm) in apical and basal dendrites in LII/LIII pyramidal neurons for both controls and mutants from the same litters (for ΔNp73$^{cre/+}$;Bax$^{lox/lox}$ apical and basal dendrites: n = 8 for

*Figure 4 continued on next page*

*Figure 4 continued*

controls and n = 12 for mutants at P23-28, p=0.012 for apical dendrites and p=0.0014 for basal dendrites; for ΔNp73$^{cre/+}$;R26$^{Kir2.1/+}$ apical dendrites: n = 10 for controls and n = 6 for mutants at P23-P29, p=0.166; basal dendrites: n = 9 for controls and n = 7 for mutants, p=0.652; Mann-Whitney U Test). Scale bar represents 5 µm. (E, G) Pyramidal neurons recorded in voltage-clamp at −70 mV and 0 mV in control at P26 (E, left) and P24 (G, left) and in a ΔNp73$^{cre/+}$;Bax$^{lox/lox}$ mutant at P23 (E, right) and a ΔNp73$^{cre/+}$;R26$^{Kir2.1/+}$ mutant at P28 (G, right) during the extracellular stimulation of LII/III fibers as indicated (E, inset). Stimulation artefacts were blanked for visibility. The stimulation time is indicated (arrowheads). (F, H) Plots of E/I ratio calculated from eEPSCs and eIPSCs in controls and ΔNp73$^{cre/+}$;Bax$^{lox/lox}$ mutants (F) and ΔNp73$^{cre/+}$; R26$^{Kir2.1/+}$ mutants (H) (for ΔNp73$^{cre/+}$;Bax$^{lox/lox}$: n = 10 for controls and n = 14 for mutants, p=0.031, Student T Test; for ΔNp73$^{cre/+}$;R26$^{Kir2.1/+}$: n = 8 for controls and n = 7 for mutants, p=0.612; Mann-Whitney U Test). Data used for quantitative analyses as well as the numerical data that are represented in graphs are available in *Figure 4—figure supplement 1—source data 1*.

The online version of this article includes the following source data and figure supplement(s) for figure 4:

**Figure supplement 1.** Evoked and spontaneous EPSCs and IPSCs of LII/LIII pyramidal neurons in ΔNp73$^{cre/+}$;Bax$^{lox/lox}$ and ΔNp73$^{cre/+}$;R26$^{Kir2.1/+}$ mutants.

**Figure supplement 1—source data 1.** Spine densities, evoked and spontaneous PSCs in LII/III pyramidal neurons in both Bax and Kir2.1 models.

To test whether these morphological modifications in ΔNp73$^{cre/+}$;Bax$^{lox/lox}$ mutants is accompanied by modifications in excitatory synaptic inputs, we performed whole-cell recordings of upper layer pyramidal neurons during the extracellular stimulation of LII/III fibers. First, the membrane potential of recorded cells was maintained at −70 mV or 0 mV to respectively record evoked excitatory (eEPSCs) and inhibitory (eIPSCs) postsynaptic currents. Pyramidal cells in ΔNp73$^{cre/+}$;Bax$^{lox/lox}$ mutants showed a significant increase in the mean amplitude of eEPSCs, while that of eIPSCs remained unchanged compared to controls (*Figure 4E*, *Figure 4—figure supplement 1A*). This modification is highlighted by a significant increase in the E/I ratio (*Figure 4F*). Together with the increased spine density, these data strongly suggest that pyramidal neurons have enhanced excitatory synaptic inputs. To corroborate this possibility, we then analyzed the spontaneous EPSCs (sEPSCs) of recorded pyramidal neurons. As expected for an increased number of inputs, the sEPSC frequency was significantly higher in ΔNp73$^{cre/+}$;Bax$^{lox/lox}$ mutants with respect to controls while the mean sEPSC amplitude remained unchanged (*Figure 4—figure supplement 1C-D*). When the same experiments were performed in ΔNp73$^{cre/+}$;R26$^{Kir2.1/+}$ mutants, no differences were observed either in the amplitudes of both eEPSCs and eIPSCs or in the E/I ratio (*Figure 4G–H*, *Figure 4—figure supplement 1B*). In line with this, changes were neither observed in the frequency of sEPSCs of these mutants (*Figure 4—figure supplement 1E*), showing that defects in the synaptic activity of pyramidal neurons is dependent on the intrinsic activity of rescued CRs. To test whether the effect on pyramidal cells could be due to a direct action of CR activity on excitatory circuits, we produced ΔNp73$^{cre/+}$;Bax$^{lox/lox}$; ChR2$^{lox/+}$ mutant mice to photoactivate rescued CRs while recording neuronal network activity. After defining an efficient photostimulation protocol for reliably eliciting action potentials on recorded CRs (*Figure 4—figure supplement 1F*), we combined photostimulation with of Layer I interneuron recordings in whole-cell configuration and/or local field potentials (LFPs) in different layers (*Figure 4—figure supplement 1G*). We could never detect light-evoked responses during patch-clamp or extracellular recordings, even in the presence of 0 mM Mg$^{2+}$, 3 mM Ca$^{2+}$ and 50 µM of the potassium channel blocker 4AP, a treatment that renders all neurons more excitable (*Figure 4—figure supplement 1G*). Although these results do not rule out that rescued CRs contact other neurons through *bona fide* glutamatergic synapses, the low proportion of these cells compared to pyramidal neurons and the lack of electrical extracellular responses during their sustained light stimulation suggest that direct synaptic inputs from CRs cannot account for the robust morphological and functional changes induced in pyramidal cells by the aberrant CR survival.

Overall, these experiments show that the survival of electrically-active SE-CRs increases the excitatory inputs to upper pyramidal neurons, possibly through a non-glutamatergic mechanism, thereby generating a E/I imbalance and functional changes in circuit wiring.

## Discussion

Our results show that the elimination of specific subsets of CRs, SE-CRs, is activity-dependent and that this process is essential for proper cortical wiring. Indeed, the persistence of SE-CRs beyond their normal phase of elimination triggered major deficits in LII/LIII somatosensory pyramidal

neurons. Not only the analyzed neurons displayed an increased dendritic arborization and spine density, but they also consistently showed enhanced excitatory inputs leading to a functional E/I imbalance. Remarkably, these anatomical and electrophysiological deficits all relied on the fact that persistent CRs were electrically-active. Our study thus demonstrates that activity is required to eliminate SE-CRs, whose survival would otherwise perturb cortical wiring in an activity-dependent manner. Taken together, it reveals an elegant interplay between transient CRs and neuronal activity in the construction of functional cortical excitatory circuits.

## CR subtype-specific pathways in programmed cell death

Activity was reported to promote survival of both glutamatergic and GABAergic neurons in the neocortex and in general in the nervous system ( *Blanquie et al., 2017a*; *Causeret et al., 2018*; *Wong and Marín, 2019*), providing a mean to integrate around 70% of neurons into functional circuits. CRs, which completely undergo programmed cell death in the cerebral cortex (*Ledonne et al., 2016*; *Causeret et al., 2018*), have been previously proposed to behave differently. These neurons display 'immature' features such as a depolarized resting potential and a very high input resistance (*Kirischuk et al., 2014*). Especially, GABA is depolarizing in these cells due to their elevated intracellular chloride concentration resulting from the activity of the chloride inward transporter NKCC1 in the absence of expression of chloride outward transporter KCC2 (*Mienville, 1998*; *Achilles et al., 2007*; *Pozas et al., 2008*). Interestingly, the pharmacological blockade of NKCC1 in cell cultures or the genetic ablation of this transporter in vivo promotes the survival of a CR population, probably by preventing $GABA_A$ receptor-mediated depolarization (*Blanquie et al., 2017b*). It must be considered, however, that the depolarizing effect of GABA will depend most probably on the levels of neuronal activity since high activity levels attenuate the $GABA_A$ receptor-mediated excitatory drive in CRs (*Kolbaev et al., 2011*). Here, we confirmed that CRs receive exclusively functional GABAergic synaptic inputs and express low levels of KCC2 in the second postnatal week, that is during the period of massive cell death. Our findings raise the question of the identity of possible GABAergic neurons that regulate CR subtype elimination. CRs receive GABAergic synaptic inputs from different sources, including local interneurons of Layer I, the underlying layers of the neocortex and the subplate, a transient cortical structure absent in the fourth postnatal week, as well as the zona incerta (*Kirmse et al., 2007*; *Myakhar et al., 2011*; *Kirischuk et al., 2014*; *Chen and Kriegstein, 2015*; *Sun et al., 2019*). A related issue is how GABAergic inputs to CRs might change overtime, since we found that rescued CRs receive less inputs than at earlier stages. One interesting possibility is that rescued CRs in Layer I might lose GABAergic innervation from transient or distant populations (i.e. subplate and zona incerta) during maturation of neuronal networks, restricting their connectivity with more local neocortical inputs. Further investigation is needed to determine the different interneuron subtypes impinging on CRs in immature cortical circuits and after their aberrant survival in adults.

Using conditional Kir2.1 expression in a large subpopulation of CRs, we unequivocally show that only a specific subset of CRs, SE-CRs, dies in an activity-dependent manner in vivo. Since CRs are highly hyperpolarized in this mouse model, a plausible explanation is that GABAergic inputs cannot exert their depolarizing effect as in normal conditions, thereby preventing cell death. In this context, it is also tempting to hypothesize that, as shown in other systems, neuronal activity via intracellular calcium signals could trigger apoptosis, as reported during excitotoxicity (*Blanquie et al., 2017a*). The timing of SE-CRs death, namely the second postnatal week, corresponds to a major switch in cortical activity (*Luhmann and Khazipov, 2018*) and in GABAergic circuits (*Cossart, 2011*), raising the possibility that the elimination of CRs is part of a more global activity-dependent remodeling of cortical circuits. Irrespective of the underlying mechanism, surviving Kir2.1-expressing CRs displayed a relatively normal morphology, expressed Reln and received GABAergic synaptic inputs. Remarkably, SE-CRs is also the subpopulation undergoing a Bax-dependent apoptosis. Quantification of CRs that persist in both models suggests that a vast majority of SE-CRs survives. Indeed, in both deletion of *Bax* (*Ledonne et al., 2016*) or over-expression of Kir2.1 (this manuscript) models, a five-fold increase in CR numbers, corresponding to approximately 30% of the initial pool at P7, is detected when using the ΔNp73[Cre] line, in contrast to none when using the hem-specific Wnt3a[Cre] line. Since hem-CRs constitute about 70% of the population targeted by the ΔNp73[Cre] line (*Bielle et al., 2005*; *Yoshida et al., 2006*; *Tissir et al., 2009*), our findings support that the 30% of rescued CRs in ΔNp73[Cre] corresponds to a large fraction, if not all, of the SE-CR population. Together, these results demonstrate that hem-derived CRs die in a *Bax*- and activity-independent

manner, in contrast to SE-CRs that survive in Bax and Kir2.1 mutants. Hence, our work reveals that subpopulations of CRs are eliminated by very distinct mechanisms. Interestingly, hippocampal CRs, which mostly derive from the cortical hem (*Louvi et al., 2007*) display a delayed death which seems to occur independently of the apoptotic-specific Caspase-3 activity (*Anstötz et al., 2016*; *Anstötz et al., 2018*). Together, these data support the notion that CR subtypes are intrinsically different in the mechanism determining their demise and argues in favor of complex yet unappreciated subtype-specific pathways leading ultimately to cell death.

## CRs aberrant survival perturbs the morphology and connectivity of upper layer neurons in an activity-dependent manner

In this work, we have demonstrated that SE-CRs persistence in mice has a strong effect on LII/III pyramidal neuron morphology and excitatory circuits. Notably, in the $\Delta Np73^{cre/+}$;Bax$^{lox/lox}$ mutants, we observed an impact on both apical and basal dendrites. While the effect on apical dendrites could be direct via local surviving SE-CRs, the impact on basal ones might be circuit-mediated since excitatory entries in apical dendritic tufts were shown to modify basal dendrite synaptic plasticity (*Williams and Holtmaat, 2019*). Moreover, SE-CRs survival leads in $\Delta Np73^{cre/+}$;Bax$^{lox/lox}$ mutants to increased synaptic density with major functional consequences on the E/I ratio. Conversely, recent work showed that aberrant reduction of CRs during development triggers decreased apical dendritic tufts and dendritic spine density of LII/III pyramidal neurons accompanied by a reduction in the E/I ratio (*de Frutos et al., 2016*). Taken together our findings reveal that the proper balance of CRs constitutes an essential, yet underappreciated, regulator of LII/III pyramidal neuron morphology and wiring.

Importantly, we found in both Bax and Kir models that surviving CRs are similarly kept embedded into functional networks. Interestingly, rescued CRs are also solely innervated by GABA$_A$ receptor-mediated synapses like their younger control counterparts. Although GABAergic synaptic connectivity often increases during postnatal development (*Pangratz-Fuehrer and Hestrin, 2011*), some cells may display transient connections that disappear after the second postnatal week, thereby accounting for the reduced connectivity observed in the two models. In contrast, since the effects on pyramidal neurons and cortical excitability are only found in *Bax* mutants, our study reveals that the inappropriate survival of SE-CRs drives an abnormal cortical wiring via an activity-dependent mechanism. However, CRs appear to act on upper layer pyramidal neurons *via* partially distinct mechanisms at different time points during development. Indeed, Kir2.1 expressing mice appear largely similar to controls, suggesting that the impact of reducing CR density on LII/III apical dendrites (*de Frutos et al., 2016*) does not rely exclusively on the intrinsic excitability of CRs. Nevertheless, the mechanism by which rescued CRs induce morphological and functional changes on pyramidal neurons remains unresolved. The lack of response observed during our optogenetic experiments suggest that these changes may not depend on a direct CR excitatory synaptic input onto principal neurons. This is in line with recent data obtained with experiments performed in the hippocampus using ChR2 activation and paired-recordings in the third postnatal week, where CRs are still present in high density (*Quattrocolo and Maccaferri, 2014*; *Anstötz et al., 2016*). Indeed, only very few pyramidal cells could be detected as an output of CR cells. Further experiments will be required to determine the relative roles of CR-secreted factors versus circuit-mediated effects onto apical and basal dendrites. Regardless the mechanism and since excitatory entries onto apical dendrites are emerging as major actors in sensory gating, cortical integration and reward (*Keller and Mrsic-Flogel, 2018*; *Khan and Hofer, 2018*; *Lacefield et al., 2019*; *Williams and Holtmaat, 2019*; *Zhang and Bruno, 2019*), our findings highlight the importance of a transient cell population in the emergence of functional circuits as well as the deleterious effects of their abnormal demise.

Our study thus shows that the elimination of SE-CRs in the somatosensory cortex is required for proper morphology and wiring of LII/III pyramidal neurons and reveals a remarkable interplay between activity, the elimination of transient cells and cortical wiring. Indeed, activity, likely driven by cortical maturation, regulates the elimination of transient CRs that would otherwise perturb upper layer wiring. Notably, CRs persistence has been described in human pathological conditions, often associated with epilepsy. Our work thus not only provides novel insights onto normal wiring of upper layers, but also addresses the functional consequences of incomplete CR removal with major relevance for neurodevelopmental diseases, such as autism spectrum disorder, schizophrenia or epilepsy.

# Materials and methods

## Key resources table

| Reagent type (species) or resource | Designation | Source or reference | Identifiers | Additional information |
|---|---|---|---|---|
| Strain *Mus musculus* (males and female) | C57BL6J | Janvier | | |
| *Mus musculus* (males and female) | ΔNp73$^{CreIRESGFP}$ | *Tissir et al., 2009* | ΔNp73$^{Cre}$ | |
| *Mus musculus* (males and female) | Wnt3a$^{Cre}$ | *Yoshida et al., 2006* | Wnt3a$^{Cre}$ | |
| *Mus musculus* (males and female) | Tau$^{loxP-stop-loxP-MARCKSeGFP-IRES-nlslacZ}$ | *Hippenmeyer et al., 2005* | Tau$^{GFP}$ | |
| *Mus musculus* (males and female) | Bax$^{tm2Sjk}$. Bak1$^{tm1Thsn}$/J | *Takeuchi et al., 2005* | Bax$^{lox/lox}$ | |
| *Mus musculus* (males and female) | ROSA26$^{loxP-stop-loxP-Tomato}$ | *Madisen et al., 2010* | R26$^{mT}$ | |
| *Mus musculus* (males and female) | ROSA26$^{loxP-stop-loxP-Kcnj2-cherry/+}$ | *Moreno-Juan et al., 2017* | R26$^{Kir2.1/+}$ | |
| *Mus musculus* (males and female) | Ai32(RCL-ChR2 (H134R)/EYFP | https://www.jax.org/strain/012569 | ChR2$^{lox}$ | |
| Antibody | rabbit polyclonal anti-DsRed | Takara | RRID:AB_10013483 | IF(1:500) |
| Antibody | mouse monoclonal anti-Reelin | Merck Millipore | RRID:AB_565117 | IF(1:300) |
| Antibody | mouse monoclonal anti-Gephyrin | Synaptic systems | RRID:AB_2619837 | IF(1:250) |
| Antibody | rabbit polyclonal anti-GAD65/67 | Merck | RRID: AB_2278725 | IF(1:250) |
| Antibody | guinea-pig anti-KCC2 | D Ng and S Morton TM Jessell's lab | | IF(1:4000) |
| Antibody | donkey anti-mouse Alexa-488 | Jackson Immuno Research Laboratories | RRID:AB_2340846 | IF(1:800) |
| Antibody | donkey anti-rabbit Cy3 | Jackson ImmunoResearch Laboratories | RRID:AB_2307443 | IF(1:800) |
| Antibody | donkey anti-rabbit Alexa-647 | Molecular Probes | RRID:AB_2536183 | IF(1:500) |
| Antibody | donkey anti-chick Alexa-488 | Jackson ImmunoResearch Laboratories | RRID:AB_2340375 | IF(1:1000) |
| Antibody | donkey anti-mouse Alexa-555 | Molecular Probes | RRID:AB_2536180 | IF(1:1000) |
| Antibody | goat anti-guinea pig Alexa-555 | Molecular Probes | RRID:AB_2535856 | IF(1:1000) |
| Antibody | DAPI (4', 6-diamidino-2-phenylindole) | Invitrogen Molecular Probes | RRID:AB_2629482 | IF(1:2000) |
| Antibody | DyLight 488 streptavidin | Vector Labs | SP-4488 | |
| Sequence-based reagent | CRE genotyping 188 f 167 r | This paper | PCR primers | 188 f: TGA TGG ACA TGT TCA GGG ATC 167 r: GAA ATC AGT GCG TTC GAA CGC TAG A |

*Continued on next page*

*Continued*

| Reagent type (species) or resource | Designation | Source or reference | Identifiers | Additional information |
|---|---|---|---|---|
| Sequence-based reagent | R26$^{Kir2.1/+}$ genotyping AAY101 AAY103 SD297 | | PCR primers | AAY101: AAAGTCGCTCTGAGTTGTTAT (Rosa26 forward WT) AAY103: GGGAGCGGGAGAAATGGATATG (Rosa26 reverse WT) SD297: GGCCATTTACCGTAAGTTATG (CAG promoter reverse) |
| Chemical compound, drug | Paraformaldehyde | Sigma-Aldrich | CAT:P6148 | |
| Chemical compound, drug | Triton 100X | Eurobio | CAT:GAUTTR00-07 | |
| Chemical compound, drug | SR95531 | Abcam | Ab120042 | |
| Chemical compound, drug | 4-AP | Sigma Aldrich | A-0152 | |
| Software, algorithm | IMARIS software 8.4. | IMARIS | RRID:SCR_007370 | |
| Software, algorithm | GraphPad Prism 7.0 | GraphPad Software | RRID:SCR_000306 | |
| Software, algorithm | ImageJ/FIJI | NIH | RRID:SCR_002285 | |
| Software, algorithm | Adobe Photoshop CS6 | Adobe Systems | RRID:SCR_014199 | |
| Software, algorithm | pClamp10.1 | Molecular Devices | RRID:SCR_011323 | |
| Software, algorithm | IGOR Pro 6.0 | Wavemetrics | RRID:SCR_000325 | |
| Software, algorithm | NeuroMatic | Wavemetrics | RRID:SCR_004186 | |

## Animals

ΔNp73$^{CreIRESGFP}$(ΔNp73$^{Cre}$) (*Tissir et al., 2009*), Wnt3a$^{Cre}$ (*Yoshida et al., 2006*), ROSA26$^{loxP-stop-loxP-Tomato}$(R26$^{mT}$) (*Madisen et al., 2010*), Tau$^{loxP-stop-loxP-MARCKSeGFP-IRES-nlslacZ}$ (Tau$^{GFP}$) (*Hippenmeyer et al., 2005*) and ChR2$^{lox}$ (Ai32(RCL-ChR2(H134R)/EYFP) (https://www.jax.org/strain/012569)) transgenic mice were kept in a C57BL/6J background. The Bax$^{tm2Sjk}$;Bak1$^{tm1Thsn}$/J line (*Takeuchi et al., 2005*) harboring the floxed *Bax* and the *Bak* knock-out alleles was purchased from the Jackson laboratory as mixed B6;129. ΔNp73$^{Cre}$ and Wnt3a$^{Cre}$ lines were crossed with the R26$^{mT}$ or Tau$^{GFP}$ reporter lines to permanently label CR subtypes. ΔNp73$^{Cre}$ line was crossed to the Bax$^{tm2Sjk}$;Bak1$^{tm1Thsn}$/J line (Bax$^{lox/lox}$) to inactivate Bax function in specific CR subtypes. The conditional knock-out also harbored a reporter allele R26$^{mT}$ in order to trace the neurons in which recombination had occurred. ΔNp73$^{Cre}$ and Wnt3a$^{Cre}$ lines were crossed to ROSA26 $^{loxP-stop-loxP- Kcnj2-cherry/+}$ (R26$^{Kir2.1/+}$) (*Moreno-Juan et al., 2017*) to overexpress the *Kcnj2* gene, which encodes Kir2.1, in specific CR subpopulations. Controls used were littermates heterozygous *Bax* for the Bax model and ΔNp73$^{+/+}$;R26$^{Kir2.1/+}$ for the Kir2.1 model. Animals were genotyped by PCR using primers specific for the different alleles. All animals were handled in strict accordance with good animal practice as defined by the national animal welfare bodies, and all mouse work was approved by the Veterinary Services of Paris (Authorization number: 75–1454) and by the Animal Experimentation Ethical Committee Buffon (CEEA-40) (Reference: CEB-34–2012) and by the Animal Experimentation Ethical Committee Darwin (Reference: 02224.02).

## Tissue preparation and immunohistochemistry

For staging of animals, the birth date was considered as postnatal day 0 (P0). Animals were anesthetized with Isoflurane and intracardially perfused with 4% paraformaldehyde (PFA) in 0.1 M PBS, pH 7.4 and post-fixed over-night in 4% PFA at 4°C. Brain were embedded in 3.5% agarose and sectioned in 70 μm free-floating slices at all stages in *Figure 1* and *Figure 1—figure supplement 1*. For

*Figure 2—figure supplement 1E*, brains were cryoprotected and sectioned in 50 µm free-floating slices. Immunostaining was performed as previously described (*Bielle et al., 2005*; *Griveau et al., 2010*; *de Frutos et al., 2016*). Primary antibodies used for immunohistochemistry were: mouse anti-Reelin (MAB5364, Millipore 1:300), rabbit anti-DsRed (Takara 632496, 1:500), mouse anti-Gephyrin (147 011, Synaptic Systems, 1:250), Rabbit anti-GAD65/67 (AB1511, Merck, 1:250), guinea pig anti-KCC2 (Gift of S.Morton and D.Ng, 1:4000). Secondary antibodies used against primary antibodies were: donkey anti-mouse Alexa-488 (Jackson ImmunoResearch Laboratories, 1:800), donkey anti-rabbit Cy3 (Jackson ImmunoResearch Laboratories, 1:800), donkey anti-chick Alexa-488 (Jackson ImmunoResearch Laboratories, 1:1000), donkey anti-mouse Alexa-555 (A-31570, Molecular Probes, 1:1000), donkey anti-rabbit Alexa-647 (A31573, Molecular Probes, 1:500), goat anti-guinea pig Alexa-555 (A-21435, Molecular Probes, 1:1000). Hoechst (Sigma-Aldrich 33342, 1:1000) and DAPI (D1306, ThermoFisher Scientific, 1:2000) were used for fluorescent nuclear counterstaining the tissue and mounting was done in Vectashield (Vector Labs).

## Image acquisition and cell countings

Immunofluorescence images were acquired using a confocal microscope (Leica TCS SP5), except for anti-KCC2, Gephyrin and GAD65/67 (*Figure 2—figure supplement 1E, f*) that were acquired on a LEICA SP8 confocal microscope with 93X objective and 2.5 digital zoom (a single optical plane or around 100 z-stacks of 0.07 µm respectively). DsRed$^+$ neurons, detected by immunofluorescence, were counted using the ImageJ software, in the somatosensory barrel cortex (S1) for each age and genotype. For each section, the density of CRs (DsRed$^+$ CRs/mm$^3$) was calculated taking into account the thickness of the section and the surface of Layer I, measured using ImageJ software.

## Acute slice preparation, electrophysiology and photostimulation

Acute coronal slices (300 µm) of the neocortex were obtained from ΔNp73$^{cre/+}$;Bax$^{lox/lox}$ and ΔNp73$^{cre/+}$;R26$^{Kir2.1/+}$ mutants. Excitation light to visualize the tdTomato or Cherry fluorescent proteins was provided by a green Optoled Light Source (Cairn Research, UK) and images were collected with an iXon+ 14-bit digital camera (Andor Technology, UK), as previously described (*Orduz et al., 2015*). Patch-clamp recordings were performed at RT using an extracellular solution containing (in mM): 126 NaCl, 2.5 KCl, 1.25 NaH$_2$PO$_4$, 26 NaHCO$_3$, 20 glucose, five pyruvate, 2 CaCl$_2$ and 1 MgCl$_2$ (95% O$_2$, 5% CO$_2$). Fluorescent CRs were recorded at P13-17 and P24-29 with different intracellular solutions according to the experiment and containing (in mM): either 130 K-Gluconate (K-Glu) or 130 KCl, 0.1 EGTA, 0.5 CaCl$_2$, 2 MgCl$_2$, 10 HEPES, 2 Na$_2$-ATP, 0.2 Na-GTP and 10 Na$_2$-phosphocreatine and 5.4 mM biocytin (pH ≈ 7.3). When using a KGlu-based intracellular solution in whole-cell configuration, potentials were corrected for a junction potential of −10 mV. Recordings were made without series resistance (R$_s$) compensation; R$_s$ was monitored during recordings and cells showing a change of more than 20% in R$_s$ were discarded. To evaluate the effect of SR95351 (10 µM; Abcam, Cambridge, UK), drug perfusion reached a steady state at 3 min in the recording chamber. This time was respected before quantification of either spontaneous or evoked PSCs. To test for the presence of eEPSCs mediated by NMDARs receptors at hyperpolarized potentials, the extracellular concentration of MgCl$_2$ was replaced by CaCl$_2$ in order to relieve the Mg$^{2+}$ block of these receptors.

Photostimulation of fluorescent ChR2-expressing rescued CRs recorded in whole-cell configuration was obtained by triggering light trains with a blue LED (470 nm, 1 ms pulses; Optoled Light Source, Cairn Research, UK). Light trains of 2, 5, 10 and 20 Hz during 10 s or 30 s were applied to define the optimal frequency inducing an effective activation of recorded ChR2-expressing rescued CRs. For each frequency, we calculated the number of spikes (Ns) with respect to the number of light pulses (NLP) and determined the percentage of success as [Ns/NLP] x 100. To test the effect of rescued CRs activation, we performed extracellular recordings and patch-clamp recordings of Layer I interneurons while stimulating with light trains (5 Hz, 10 s or 30 s). Extracellular recordings were recorded with a patch pipette filled with extracellular solution. In a set of experiments, the extracellular solution contained 0 mM Mg$^{2+}$, 3 mM Ca$^{2+}$ and 4AP for more than 5 min at least.

Recordings were obtained using Multiclamp 700B and pClamp10.1 (Molecular Devices), filtered at 4 kHz and digitized at 20 kHz. Digitized data were analyzed off-line using Neuromatic within IGOR Pro 6.0 environment (Wavemetrics, USA) (*Rothman and Silver, 2018*). Extracellular

stimulations were performed using a monopolar electrode (glass pipette) placed in Layer I for CRs and Layer II/III for pyramidal neurons (20–99 V, 100 μs stimulations each 8–12 s; Iso-Stim 01D, npi electronic GmbH, Tamm, Germany). Spontaneous postsynaptic currents were detected with a threshold of 2 times the noise standard deviation during a time window of 3 min for CRs and 1.5 min for pyramidal cells. The Vm was estimated in current-clamp mode as soon as the whole-cell configuration was established. The analysis of Rin, action potential amplitudes and duration was performed during pulses of 800 ms in current-clamp configuration from −80 mV during increasing steps of 5 pA as previously described (*Ledonne et al., 2016*).

## Morphological analyses

For morphological analysis, CRs and layer II/III pyramidal cells were loaded with biocytin through patch pipette during whole-cell recordings. The slices were fixed 2 hr in 4% paraformaldehyde at 4° C, rinsed three times in PBS for 10 min, and incubated with 1% triton X-100% and 2% BSA during 1 hr. Then, they were washed three times in PBS and incubated in DyLight 488 streptavidin (Vector Labs, Burlingame, USA) for 2 hr. Successfully labeled CRs and Layer II/III pyramidal cells were visualized either using a LEICA SP5 or SP8 confocal microscope with a 40X objective and a 1.3 digital zoom. Around 250 optical sections of 0.6 μm were necessary to image the whole dendritic tree of each neuron. For dendritic spines, images of apical dendrites in Layer I and basal dendrites in Layer II/III were acquired with a 63X objective (100 optical sections of 0.25 μm each). For apical dendrites, a terminal ramification in Layer I was acquired, while for basal dendrites a horizontal dendrite in Layer II/III was acquired approximately at 60 μm distance from the soma after the first ramification. 3D reconstruction was performed using the IMARIS software 8.4. Statistical analyses were performed based on the data given by IMARIS in apical and basal dendrites separately. Counting of spines was performed manually using ImageJ software on black and white maximum projections on two different segments approximately 50 μm long for each image. For morphological analysis of CRs, statistical analyses were performed based on the values calculated by image analyses using the IMARIS software (soma diameter and filament length).

## Statistical analysis

All data were expressed as mean ± SEM. A P-value less than 0.05 was considered significant. For statistical groups larger than 7, we performed a D'Agostino-Pearson normality test. According to the data structure, two-group comparisons were performed using two-tailed unpaired Student T test or Mann-Whitney U Test. Bonferroni multiple comparisons were used as post-hoc test following one-way or two-way ANOVA or the non-parametric Kruskal-Wallis Tests. For small statistical groups (less than 7), we systematically performed non-parametric tests (Mann-Whitney U Test or Kruskal-Wallis Tests with Dunn's correction). A reconstructed pyramidal neuron displaying an exceptional basal dendrite projecting to Layer I was submitted to Grubbs test and excluded as an extreme outlier. Statistics and plotting were performed using GraphPad Prism 7.00 (GraphPad Software Inc, USA). *p<0.05, **p<0.01, ***p<0.001.

## Acknowledgements

The authors apologize for not having been able to cite the work of many contributors to the field. We wish to thank Q Dholandre, L Vigier, M Keita, D Souchet, C Auger, A Delecourt, E Touzalin, D Valera and C Le Moal, for help with the mouse colonies and genotyping, L Danglot, S Morton and D Ng for providing antibodies and A Ben Abdelkrim for help with statistical analyses, as well as members of the Pierani, Angulo and Garel laboratories for helpful discussions and critical reading of the manuscript. We acknowledge the ImagoSeine facility, member of the France BioImaging infrastructure supported by the French National Research Agency (ANR-10-INSB-04, 'Investments for the future') for help with confocal microscopy, Animalliance for technical assistance and animal care. We thank the IBENS Imaging Facility (France BioImaging, supported by ANR-10-INBS-04, ANR-10-LABX-54 MEMO LIFE and ANR-11-IDEX-000–02 PSL* Research University, 'Investments for the future'). We acknowledge NeurImag facility of IPNP. AP and MCA are CNRS (Centre National de la Recherche Scientifique) Investigators, SG is an Inserm researcher, and all member Teams of the École des Neurosciences de Paris Ile-de-France (ENP), EC is a University Paris Diderot Lecturer, MR and IG are supported by fellowships from the French Ministry of Research, CH by a postdoctoral

fellowship from Fondation pour l'aide à la recherche sur la Sclérose en Plaques (ARSEP). This work was supported by grants from the ANR-15-CE16-0003-01, FRM («Equipe FRM DEQ20130326521») to AP and State funding from the Agence Nationale de la Recherche under 'Investissements d'avenir' program (ANR-10-IAHU-01) to the Imagine Institute, Fondation pour la Recherche Médicale (FRM, «Equipe FRM DEQ20150331681») to MCA, grants from INSERM, CNRS and the ERC Consolidator Grant NImO 616080 to SG, ERC Consolidator Grant (ERC-2014-CoG-647012) and the Spanish Ministry of Science, Innovation and Universities (BFU2015-64432-R) to GL-B.

## Additional information

### Funding

| Funder | Grant reference number | Author |
|---|---|---|
| Agence Nationale de la Recherche | ANR-15-CE16-0003-01 | Alessandra Pierani<br>Maria Cecilia Angulo<br>Sonia Garel |
| Fondation pour la Recherche Médicale | Equipe (DEQ20130326521) | Alessandra Pierani |
| Fondation pour la Recherche Médicale | Equipe (DEQ20150331681) | Maria Cecilia Angulo |
| European Commission | ERC-2013-CoG-616080 | Sonia Garel |
| Ministry of Higher Education, Research and Innovation | Fellowship | Martina Riva<br>Ioana Genescu |
| Ministry of Science, Innovation and Universities | BFU2015-64432-R | Guillermina López-Bendito |
| European Commission | ERC-2014-CoG-647012 | Guillermina López-Bendito |
| Fondation pour l'Aide à la Recherche sur la Sclérose en Plaques | Postdoctoral fellowship | Chloé Habermacher |

The funders had no role in study design, data collection and interpretation, or the decision to submit the work for publication.

### Author contributions

Martina Riva, Ioana Genescu, Chloé Habermacher, David Orduz, Eva Coppola, Conceptualization, Data curation, Formal analysis, Validation, Visualization, Methodology, Writing—original draft, Writing—review and editing; Fanny Ledonne, Data curation, Formal analysis, Supervision, Validation, Visualization, Methodology; Filippo M Rijli, Guillermina López-Bendito, Resources; Sonia Garel, Alessandra Pierani, Conceptualization, Resources, Formal analysis, Supervision, Funding acquisition, Validation, Investigation, Visualization, Methodology, Writing—original draft, Project administration, Writing—review and editing; Maria Cecilia Angulo, Conceptualization, Formal analysis, Supervision, Funding acquisition, Validation, Investigation, Visualization, Methodology, Writing—original draft, Project administration, Writing—review and editing

### Author ORCIDs

David Orduz https://orcid.org/0000-0002-4198-2691
Filippo M Rijli http://orcid.org/0000-0003-0515-0182
Sonia Garel https://orcid.org/0000-0003-2984-3645
Maria Cecilia Angulo https://orcid.org/0000-0002-0758-0496
Alessandra Pierani https://orcid.org/0000-0002-4872-4791

### Ethics

Animal experimentation: All animals were handled in strict accordance with good animal practice as defined by the national animal welfare bodies, and all mouse work was approved by the Veterinary Services of Paris (Authorization number: 75-1454) and by the Animal Experimentation Ethical

Committee Buffon (CEEA-40) (Reference: CEB-34-2012) and by the Animal Experimentation Ethical Committee Darwin (Reference: 02224.02).

## Decision letter and Author response
Decision letter https://doi.org/10.7554/eLife.50503.sa1
Author response https://doi.org/10.7554/eLife.50503.sa2

## Additional files
### Supplementary files
- Transparent reporting form

### Data availability
All data generated or analysed during this study are included in the manuscript and supporting files. The Source data file contains all the data presented in the figures (1 sheet per Figure).

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
