## [Decision Letter]

**Acceptance summary:**

In this paper, you have provided novel information on the role of the subset of Cajal-Retzius (CR) neurons derived from the septum and thalamic eminences in the maturation of cortical circuits. You show that upon overexpressing Kir2.1, this subpopulation of CR cells can be rescued from developmental cell death, then become functionally integrated into cortical circuits. You also nicely demonstrate that spontaneous and evoked postsynaptic currents observed in the rescued CRs can be abolished by GABA-AR antagonists, implicating GABAergic inputs in regulating CR neuronal activity. Of interest is that the CRs aberrantly survive after hyperpolarization and trigger dendritic and spine complexity and electrophysiological properties of L2/3 pyramidal neurons, leading to an increased E/I ratio. In all, your paper highlights activity-dependent programmed cell death that is required for the proper formation of cortical circuits.

**Decision letter after peer review:**

Thank you for sending your article entitled "Activity-dependent death of transient Cajal-Retzius neurons is required for functional cortical wiring" for peer review at *eLife*. Your article is being evaluated by three peer reviewers, and the evaluation is being overseen by a Reviewing Editor and Marianne Bronner as the Senior Editor.

Given the list of essential revisions, including new experiments, the editors and reviewers invite you to respond within the next two weeks with an action plan and timetable for the completion of the additional work. We plan to share your responses with the reviewers and then issue a binding recommendation.

Two of the three reviewers, and the Reviewing Editor, are enthusiastic about your work, and considered the data that Cajal-Retzius (CR) developmental cell death is regulated by neuronal activity "innovative and highly interesting", "convincing", and "providing important new insights" on the role of the subset of CR neurons derived from the septum and thalamic eminences in the maturation of cortical circuits. You demonstrate that upon overexpressing Kir2.1, this subpopulation of CR cells can be rescued from apoptosis and functionally integrated into cortical circuits. You nicely show that spontaneous and evoked postsynaptic currents observed in the rescued CRs can be abolished by GABA-AR antagonist application, and you thus implicate GABAergic inputs as regulating CR neuron activity. Of interest is that the CRs that aberrantly survive promote dendritic and spine complexity and electrophysiological properties of in L2/3 pyramidal neurons, thus leading to an increased E/I ratio. These data add to your previous study and are novel.

However, the appended reviews and off-line reviewer consultation raise several questions and suggestions made for addressing them, that would strengthen the paper. Below the requests for additional data are summarized, in order of feasibility and time required to perform the analyses.

1) Even though NKCC1 expression and lack of KCC2 in identified CR cells have been reported by single-cell PCR (Figures 4 in Achilles et al.), demonstrate that KCC2 is not expressed in CR postnatally, by immunohistochemistry.

2) Confirm the identity of local GABAergic neurons that provide these inputs to CR cells to regulate CR activity, via EM evidence for GABAergic synapses on postnatal CR as they undergo cell death (this can be done with or without associated immunohistochemistry).

3) Output from CR cells: Provide evidence for direct connections between CR cells onto L2/L3 projections neurons, with optogenetic approaches (e.g., introducing CHR2 in CR neurons and recording L2/3 neurons.

4) Further experimental analyses on the molecular mechanisms connecting neuronal activity and apoptosis requiring controlled activation of CR cells (e.g. via optogenetics, as above) and subsequent analysis of the expression profile of death and/or survival factors in single cells undergoing apoptosis. This would admittedly be a very challenging and beyond the scope of this study.

Reviewer #2:

Using mouse lines targeting CRs from different lineages and conditional genetic hyperpolarization, the authors demonstrate that apoptosis of SE-derived CRs – but not of hem-derived CRs – is prevented upon membrane hyperpolarization. It has been recently shown that SE-derived CRs, but not hem-derived CRs, undergo a Bax-dependent apoptosis. Using conditional inactivation of Bax, the authors further demonstrate that the SE-derived CRs rescued upon hyperpolarization stay embedded in the cortical network. Finally, the authors demonstrate that the rescued, electrically active CRs – but not the rescued, hyperpolarized CRs – promote dendritic complexity in LII-III pyramidal neurons, thus leading to an increased E/I ratio. Altogether, the generated data are innovative and highly interesting.

Reviewer #3:

Programmed cell death of Cajal-Retzius neurons (CR) regulates cortical circuit assembly and authors provide important new data to support this idea. CR neurons are diverse and authors used two Cre lines to study two distinct subpopulations of CR (ΔNp73-CRs and Wnt3a-derived hem-CRs). Theses Cre-driver lines are well-characterized and allow a precise cellular dissection of the developmental process of postnatal programmed cell death. Previous work from the group has revealed that ΔNp73-CRs (but not Wnt3a-derived CRs) are eliminated postnatally via Bax-dependent death. Authors follow-up on this seminal observation and provide new convincing data showing that intrinsic neuronal excitability regulates the process of CR programmed cell death. More specifically they nicely show that the density of ΔNp73-CRs is increased postnatally by selectively hyperpolarizing these cells through Kir2.1 overexpression. Following this key finding, authors provide electrophysiological data to demonstrate that ΔNp73-CRs overexpressing Kir2.1 (ΔNp73^cre/+^;R26^Kir2.1/+^) are functionally integrated in cortical circuits. Interestingly spontaneous (sPSCs) and evoked postsynaptic currents (ePSCs) observed in ΔNp73^cre/+^;R26^Kir2.1/+^ were found to be abolished by GABA-AR antagonist application, thus suggesting that GABAergic inputs are important regulators of CR neuron activity. Finally authors convincingly show that aberrant CR neuron survival affects the dendritic morphology, spine density and electrophysiological properties of L2/3 pyramidal cells. Overall, this study provides important new insights on the role of a subset of CR in the postnatal maturation of cortical circuits and show that the synaptic activity of pyramidal neurons is dependent on the intrinsic activity of rescued CRs.

My questions are mainly centered on input-output connectivity of CR neurons. Spontaneous and evoked postsynaptic currents were abolished by GABA-AR antagonist in the ΔNp73^cre/+^;R26^Kir2.1^, thus suggesting that GABAergic synaptic inputs are likely to be important regulators of the activity of postnatal CR cells. What is the possible identity of local GABAergic neurons that provide these inputs to CR cells ? Are they located in deep cortical layers (Martinotti SST+ INs ?) or locally in L1 ? This is a difficult question to address given the high diversity of GABergic interneurons. Authors could comment on this point. GABA is depolarizing in CR cells due to NKCC1 expression and absence of KCC2 expression. Has KCC2 expression been assessed postnatally during the critical period of CR cell death ? Regarding GABAergic inputs to CR: is there evidence at the EM level for GABAergic synapses on postnatal CR as they undergo cell death ? Have authors tried to find direct evidence for GABAA receptor-mediated depolarization in CR cells?

CR cells abnormally rescued from cell death affect the morphology/connectivity of L2/L3 projections neurons. Using optogenetic approaches (e.g. introducing CHR2 in CR neurons and recording L2/3 neurons), is there evidence for a direct connectivity of CR cells onto L2/L3 projections neurons ?

Reviewer #4:

Riva et al. investigated the mechanisms underlying cell death in Cajal-Retzius (CR) neurons and the role of CR cell death in circuit formation. They found that CR cell death was regulated by neuronal activity. Furthermore, they found that elimination of CR neurons regulated the morphology and functional properties of cortical neurons. Although their data is convincing and potentially interesting, novel mechanistic insights were limited, and therefore I must say that, unfortunately, their data did not provide sufficient conceptional advances.

1) The authors showed that apoptosis of ΔNp73-CR was suppressed by Kir2.1-induced hyperpolarization. The molecular mechanisms connecting neuronal activity and apoptosis are missing. Although the authors discussed a potential hypothesis, it would be much more exciting if the authors could show some experimental data supporting their hypothesis.

2) The authors demonstrated that ΔNp73^Cre/+^;Bax^lox/lox^, rather than ΔNp73^Cre/+^;R26^Kir2.1/+^, affected dendritic development, spine densities and EPSC. These are interesting observations, but the authors had already reported consistent results in their previous report (Neuron, 92, 435-448, 2016). I am afraid to say that their new data were not novel enough. If the authors could uncover some of the mechanisms of how CR neurons affect various aspects of cortical neurons, this paper would be much more attractive.

---

## [Author Response]

We are grateful to the reviewers and editors for their constructive feedback on our Short Report. While most of them highlighted the interest and novelty of our work, they also raised some concerns that we have addressed by performing additional experiments now included in an entirely novel figure (Figure 2—figure supplement 1) and Figures 1, Figure 1—figure supplement 1, Figure 3—figure supplement 1 and Figure 4—figure supplement 1. These data confirm i) low KCC2 expression in these cells and ii) the presence of GABAergic inputs both electrophysiologically and by immunohistochemistry onto genetically labeled CRs at the time of death and iii) provide evidence of lack of direct connections of CRs onto nearby neurons using optogenetics. We have modified the text accordingly in Results and Discussion and stressed the novelty of our findings. Please find below point-by-point answers and revisions.

“Your comments are well supported, and we look forward to seeing a revision addressing the 1) KCC2 IHC 2) GABA inputs to CR 3) and CR inputs to PNs. We understand that you have tried to demonstrate a direct synaptic contact between CRPNs but were unsuccessful. Although the data are negative, the reviewers welcome the inclusion of these data as a figure supplement.”

We are grateful to reviewers and the editor for their constructive feedback. We provide a revised version of our manuscript in which we address the three main points raised by the reviewers.

We have performed KCC2 immunostainings (presented in novel Figure 2—figure supplement 1) that highlight the low to undetectable expression level of the protein in CRs, identified by the *ΔNp73cre* line. As the patchy nature of the staining precludes quantification, we have further included non-CR cells labelled by this line in the hypothalamus for comparison. Together with previous publications examining RNA levels (Achilles et al., 2007; Pozas et al., 2008), our additional experiment confirms that CRs express undetectable protein levels of KCC2.

Regarding GABAergic inputs to CRs, we have performed two sets of experiments that are both presented in Figure 2—figure supplement 1. On the one hand, we have characterized by whole cell recording at P9-P11 the synaptic inputs onto genetically labeled CRs and showed that both spontaneous and evoked post-synaptic currents are solely mediated by GABAA receptors. On the other hand, we have performed immunohistochemistry with pre (GAD65/65) and post-synaptic markers (Gephryn) to visualize GABAergic synapses onto CRs. Taken together, these experiments highlight that during the time period of massive cell death (P9-P11), CRs receive exclusively GABAergic synaptic inputs.

Finally, and as discussed with the reviewers, we present in Figure 4—figure supplement 1 our attempts to identify CR outputs. We have generated *ΔNp73^cre/+^;Bax^lox/lox^;ChR2^lox//+^*mutant mice, as proposed by the reviewers. While we could establish an efficient protocol to stimulate CRs with light, as assessed by their recording, we were struck by the limited number of these cells compared to the density of pyramidal cells. We therefore decided to record local field potentials close to the light stimulation site to test whether the photoactivation of CRs was able to induce any effect on neuronal networks. In other experiments, we also patched nearby Layer I interneurons while simultaneously recording the extracellular activity with a second pipette during light train stimulations.

Unfortunately, we never detected a light-evoked response in any of these experiments, even in the presence of 0 mM Mg^2+^, 3 mM Ca^2+^ and 50 µM 4AP, a treatment that renders neurons more excitable. These findings are consistent with experiments performed in the hippocampus using ChR2 activation and paired-recordings (Quattrocolo and Maccaferri, 2014; Anstötz et al., 2016), where CRs are still present in high density during the first three postnatal weeks and only very few pyramidal cells could be detected as direct output of CRs. Although these results do not entirely preclude that cortical CRs contact Layer II/III neurons through *bona fide* synapses, they reveal that it is an extremely difficult point to assess. For these reasons, we decided not to pursue these experiments, but include them in Figure 4—figure supplement 1 and discuss their implications in the report as requested by the reviewers.

Collectively, we believe that our additional experiments significantly reinforce our claims, by providing a compelling framework on how activity might eliminate specific subpopulations of CRs and how they could impinge onto cortical development. We have also significantly rewritten the manuscript as suggested by reviewers and addressed their specific comments as detailed below.

Reviewer #3:[…] My questions are mainly centered on input-output connectivity of CR neurons. Spontaneous and evoked postsynaptic currents were abolished by GABA-AR antagonist in the ΔNp73^cre/+^;R26^Kir2.1^, thus suggesting that GABAergic synaptic inputs are likely to be important reglators of the activity of postnatal CR cells. What is the possible identity of local GABAergic neurons that provide these inputs to CR cells ? Are they located in deep cortical layers (Martinotti SST+ INs ?) or locally in L1? This is a difficult question to address given the high diversity of GABergic interneurons. Authors could comment on this point. GABA is depolarizing in CR cells due to NKCC1 expression and absence of KCC2 expression. Has KCC2 expression been assessed postnatally during the critical period of CR cell death ? Regarding GABAergic inputs to CR: is there evidence at the EM level for GABAergic synapses on postnatal CR as they undergo cell death ? Have authors tried to find direct evidence for GABAA receptor-mediated depolarization in CR cells?CR cells abnormally rescued from cell death affect the morphology/connectivity of L2/L3 projections neurons. Using optogenetic approaches (e.g. introducing CHR2 in CR neurons and recording L2/3 neurons), is there evidence for a direct connectivity of CR cells onto L2/L3 projections neurons ?

We have addressed the main points raised by reviewer 3 by adding the novel experiments now presented in Figures 2—figure supplement 1 and Figure 4—figure supplement 1.

Reviewer #4:Riva et al. investigated the mechanisms underlying cell death in Cajal-Retzius (CR) neurons and the role of CR cell death in circuit formation. They found that CR cell death was regulated by neuronal activity. Furthermore, they found that elimination of CR neurons regulated the morphology and functional properties of cortical neurons. Although their data is convincing and potentially interesting, novel mechanistic insights were limited, and therefore I must say that, unfortunately, their data did not provide sufficient conceptional advances.1) The authors showed that apoptosis of deltaNp73-CR was suppressed by Kir2.1-induced hyperpolarization. The molecular mechanisms connecting neuronal activity and apoptosis are missing. Although the authors discussed a potential hypothesis, it would be much more exciting if the authors could show some experimental data supporting their hypothesis.

While we fully agree with reviewer 4 on the importance of identifying the molecular link between neuronal activity and apoptosis, we also believe that this lies outside of the scope of our short report and deserves a full independent study on the topic.

2) The authors demonstrated that deltaNp73^Cre/+^;Bax ^lox/lox^, rather than deltaNp73^Cre/+^;R26^Kir2.1/+^, affected dendritic development, spine densities and EPSC. These are interesting observations, but the authors had already reported consistent results in their previous report (Neuron, 92, 435-448, 2016). I am afraid to say that their new data were not novel enough. If the authors could uncover some of the mechanisms of how CR neurons affect various aspects of cortical neurons, this paper would be much more attractive.

We are grateful to reviewer 4 for pointing out that our initial submission could have stressed more the novelty of our findings. We previously showed that a reduced density of CRs during normal development alters spine densities and EPSC (De Frutos et al., 2016).

Our current study addressed the mechanisms controlling the demise of CRs, as well as the impact of abnormal survival of these cells. Not only do we show that the death of a specific population of CRs receiving solely GABAergic inputs is activity-dependent, we furthermore demonstrate that the aberrant survival of such a minor population of active transient neurons is sufficient to drive an imbalance of the excitatory entries on pyramidal neurons. These findings have major implications on the activity-dependent mechanisms controlling cortical wiring in health and disease. We have now stressed the novelty of our findings throughout the Results and Discussion section.